# Cardiovascular Biomarkers: Lessons of the Past and Prospects for the Future

**DOI:** 10.3390/ijms23105680

**Published:** 2022-05-19

**Authors:** Farah Omran, Ioannis Kyrou, Faizel Osman, Ven Gee Lim, Harpal Singh Randeva, Kamaljit Chatha

**Affiliations:** 1Warwick Medical School, University of Warwick, Coventry CV4 7AL, UK; farah.omran87@hotmail.com (F.O.); kyrouj@gmail.com (I.K.); faizel.osman@uhcw.nhs.uk (F.O.); vengee.lim@uhcw.nhs.uk (V.G.L.); harpal.randeva@uhcw.nhs.uk (H.S.R.); 2Warwickshire Institute for the Study of Diabetes, Endocrinology and Metabolism (WISDEM), University Hospitals Coventry and Warwickshire NHS Trust, Coventry CV2 2DX, UK; 3Clinical Sciences Research Laboratories, University Hospitals Coventry and Warwickshire, Coventry CV2 2DX, UK; 4Centre of Applied Biological & Exercise Sciences, Faculty of Health & Life Sciences, Coventry University, Coventry CV1 5FB, UK; 5Aston Medical School, College of Health and Life Sciences, Aston University, Birmingham B4 7ET, UK; 6Laboratory of Dietetics and Quality of Life, Department of Food Science and Human Nutrition, School of Food and Nutritional Sciences, Agricultural University of Athens, 11855 Athens, Greece; 7Department of Cardiology, University Hospitals Coventry and Warwickshire NHS Trust, Coventry CV2 2DX, UK; 8Biochemistry and Immunology Department, University Hospitals Coventry and Warwickshire NHS Trust, Coventry CV2 2DX, UK

**Keywords:** cardiovascular diseases, biomarkers, future biomarkers

## Abstract

Cardiovascular diseases (CVDs) are a major healthcare burden on the population worldwide. Early detection of this disease is important in prevention and treatment to minimise morbidity and mortality. Biomarkers are a critical tool to either diagnose, screen, or provide prognostic information for pathological conditions. This review discusses the historical cardiac biomarkers used to detect these conditions, discussing their application and their limitations. Identification of new biomarkers have since replaced these and are now in use in routine clinical practice, but still do not detect all disease. Future cardiac biomarkers are showing promise in early studies, but further studies are required to show their value in improving detection of CVD above the current biomarkers. Additionally, the analytical platforms that would allow them to be adopted in healthcare are yet to be established. There is also the need to identify whether these biomarkers can be used for diagnostic, prognostic, or screening purposes, which will impact their implementation in routine clinical practice.

## 1. Cardiovascular Diseases: Overview

Cardiovascular diseases (CVDs) are the leading cause of death globally [1], including coronary and ischemic heart diseases, congestive heart failure, peripheral arterial diseases, deep vein or arterial thrombosis, pulmonary embolism (PE), and cerebrovascular diseases [2]. If detected early, many CVDs may be preventable by addressing relevant modifiable risk factors such as smoking, physical inactivity, obesity, diabetes, hypertension, and lipid disorders. Thus, it is important to have effective tools for screening, diagnosis, and prognosis. This is where biomarkers are increasingly recognized to play a key role. Aronson et al. defined a biomarker as a “biological observation that substitutes for and ideally predicts a clinically relevant endpoint or intermediate outcome that is more difficult to observe” [3]. Moreover, the National Institute of Health Consortium defined a biomarker as a “characteristic that is objectively measured and evaluated as an indicator of normal biological processes, pathogenic processes, or pharmacologic responses to a therapeutic intervention” [4,5]. Therefore, changes in biomarkers indicate physiological or pathological conditions, and may reflect responses to therapy. This wide-scope definition of biomarkers covers measurements of proteins, metabolites, and genetic factors [3]. In a broader view, imaging procedures used to identify and evaluate abnormal biological functions are also considered biomarkers [6]. Nevertheless, in 2009, the American Heart Association proposed an evaluation framework and defined criteria for how novel markers of cardiovascular risk should be evaluated in a standardized fashion before their clinical use can be recommended [7]. Depending on their role, biomarkers fall into three main classifications–namely screening, diagnostic, and prognostic biomarkers–and may have critical value in a corresponding clinical setting to evaluate risks associated with various factors related to health or disease. These factors can be the susceptibility to genetic traits or environmental factors, markers at points along the disease development, and/or progression, as well as subclinical or clinical or surrogate endpoints used in the evaluation of the safety and efficacy of therapeutic options. Moreover, focus has been increasingly placed on identifying biomarkers that can help to effectively target costlier resources (e.g., expensive treatments or invasive procedures) to the individuals/patients that would benefit most.

The present review will focus on circulating biomarkers, which can be measured in blood and are relevant to the clinical care of patients with CVDs.

## 2. Historic Overview: Early Biomarkers for CVDs

Creatine Kinase Myocardial Band (CK-MB) is one of the creatine kinase (CK) isoenzymes, representing the gold standard tests to detect and monitor cardiac injury in the 1980s [8]. CK is an intracellular enzyme that catalyses the reversible transformation of creatine and ATP to creatine phosphate and ADP, playing an important role in energy equilibrium of cells that sporadically needs high energy [9]. CK-MB is present in the myocardium, and it exists as two isoforms: namely CK-MB1 and CK-MB2, which are the plasma and tissue form, respectively [10,11]. Skeletal muscles also express CK-MB, albeit in lesser concentrations than the myocardium. [12]. As such, CK-MB is considered a cardiac marker and rises to detectable levels in the blood when there is damage to myocardium including ischemia or necrotic injury [9,13]. An increase in measured circulating CK-MB above the 99th percentile upper reference limit indicates a myocardial rather than muscular origin [14].

Elevated CK-MB values are often detected within three to four hours following the onset of acute myocardial infarction (MI), and are used to assist the diagnosis of MI [15,16] with the circulating concentration peaking within 24 h and returning to normal at 24–72 h [9,17]. Laboratory measurements of CK-MB simply illustrate the total of the isoforms CK-MB1 and CK-MB2 [18].

Apart from MI, other cardiovascular etiologies can cause elevation of CK-MB. These include myocarditis [19], pulmonary embolism [20,21], cardiac trauma [22], heart transplantation [23], chemotherapy-induced cardiotoxicity [24], and cardiac surgery [25]. However, CK-MB levels in these diseases/conditions do not add significant value to their diagnosis or the prognosis, and often cause confusion in the interpretation of the results [19,21,22,23,24,25,26,27]. Moreover, increased CK-MB levels do not specify the underlying etiology, and false-positive results can occur in a number of non-cardiovascular conditions, including end-stage renal failure [28], skeletal muscle trauma [22], muscular dystrophy [29], dermatomyositis [30], rhabdomyolysis [31], delirium tremens [32], and amyloidosis [33].

As an MI biomarker, CK-MB is now obsolete with the advent of Troponins (Tn), specifically high sensitivity Tn. As such, the CK-MB test now has limited value in the early and late diagnosis of acute MI and has been replaced by the Tn test, which is far more specific and sensitive [14,20]. However, its release kinetics can assist in diagnosing re-infarction if levels rise after initially declining following acute MI. Of note, it is reported that disparate troponin and CK-MB plasma levels are found in 28% of patients. In-hospital mortality rates are not significantly increased in patients with only CK-MB positive results compared to patients with negative values of both biomarkers [15,34]. Thus, an isolated CK-MB elevation has limited prognostic as well as diagnostic value in patients with an acute coronary syndrome (ACS). In addition, although the newer assay of evaluating CK-MB2 is more sensitive, its major disadvantage is that it is relatively labour-intensive for the laboratory, which makes using it impractical. Accordingly, the expert consensus document on the Fourth Universal Definition of MI and myocardial injury now mentions only the cardiac Tn rise above the 99th percentile upper reference limit [14].

From a historic perspective, prior to CK and CK-MB being used as cardiac biomarkers, aspartate transaminase (AST) was the first used biomarker for acute MI diagnosis. Moreover, lactate dehydrogenase (LD) and its cardiac-specific iso-enzyme (LDH-1, and to a lesser extent LDH-2) were later introduced and used in the context of acute MI. In fact, CK-MB was typically requested alongside AST and LDH as a cardiac biomarker panel in most laboratories. It is noteworthy that myoglobin also rises after acute MI and can be used as useful cardiac biomarker in the differential diagnosis of suspected acute MI. However, the absence specificity and the large occurrence of false positive results diminished the usefulness of all these biomarkers, which are now almost completely replaced by more specific cardiac biomarkers; this will be discussed in the following sections.

## 3. Current CVDs Biomarkers in Clinical Practice

### 3.1. Troponins

Cardiac troponins (cTn), or the troponin complex, are composed of three regulatory proteins, namely troponin I (cTnI), troponin T (cTnT), and troponin C (cTnC). These are integral to the cardiac muscle thin filaments (actin protein) and play a critical role in regulating cardiac muscle activity including intracellular calcium concentrations throughout the contraction/relaxation process [35]. TnC is expressed in cardiac and slow skeletal myocytes [36], whilst cardiac isoforms of TnI and TnT are translated exclusively in the heart [37,38,39,40]. The proportion of both isoforms increase during human fetal development until they are solely expressed in the heart by the ninth month after birth [37,38,39,40,41,42,43,44,45]. Thus, both TnI and TnT seem optimal for investigating heart pathologies.

Currently, the evaluation of serum levels of TnT and TnI–the highly specific troponins of cardiac myocyte damage–is one of the most valuable biomarkers in the early diagnosis of MI associated with coronary vessel and ventricular remodeling [16,46]. Accordingly, a consensus statement by the: European Society of Cardiology (ESC), American College of Cardiology Foundation (ACCF), American Heart Foundation (AHA), and World Heart Federation (WHF) in 2018 defined a MI as a rise and/or fall in cTn with at least one value above the 99th centile upper reference limit in the context of symptoms or clinical evidence of myocardial ischaemia [14,47]. Different assays can be used to measure plasma concentrations of troponins including high and standard sensitivity methods. The Joint ESC/ACCF/AHA/WHF consensus recommend the detection of high-sensitivity (hs) troponin methods at the 99th percentile of an apparently healthy reference population at <10% variability [14,48]. This approach allows earlier detection of and increased diagnosis of MI, as well as a reduction of cases with unstable angina, improved risk stratification, better medication management, and selection of early invasive vs selective invasive strategy for patients with ACS [48,49,50,51,52,53,54,55]. Nevertheless, the ESC/ACCF/AHA/WHF consensus provides no recommendations for use of the 99th percentile value for risk assessment. Risk assessment is a separate issue and hs-Tn is used in this assessment. Importantly, interpretation of cTn levels should not be performed in isolation of clinical context, but with careful clinical evaluation in emergencies such as potential acute MI because cTn levels can be detected in apparently healthy populations, especially when measured by high-sensitivity methods [56,57]. Notably, triggers of cTn leakage not related to irreversible cardiomyocyte necrosis can include elevated troponin after prolonged physical activity, inducible MI, and prolonged episodes of tachyarrhythmia, even in presumably healthy individuals [58,59]. It is, however, not clear whether elevation of troponins in such conditions are similar to cells with membrane disruption following myocardial cell death. Furthermore, it is also not known if an imbalance between oxygen demand and supply in patients with subclinical heart disease mediates such troponin secretion in apparently healthy individuals.

Overall, cTn have a short plasma half-life of around 2 h after release from injured myocardium. Of note, the released cTnT after MI exhibits discrete kinetics with two peaks; indeed, it starts to rise 3–4 h after the injury and slowly rises to a second lower peak over several days to even 2 weeks after the onset of injury [60,61,62]. cTnI is released quicker, within 1–2 h of MI [17,60,63]. As such, the pattern of cTn secretion assists with the MI diagnosis and prognosis. For instance, patients with large reperfused MI exhibit a classical biphasic time-release pattern of cTnT, while the release pattern of cTnT is different in non-reperfused MI as well as with small MIs [62,64]. It is reported that detecting the early peak may help in assessing the status of micro-vascular reperfusion, while the circulating cTn levels on day three or four represents MI size [62,65,66].

In addition, cTn are released into systemic circulation in response to cardiac myocyte injury such as in myocardial necrosis/ischemia, myocardial wall stress resulting from increased left–ventricular (LV) filling pressures, increased inflammatory cytokines, oxidative stress or catecholamine, and direct cellular destruction [62,67,68]. Thus, troponins have also gained a value as biomarkers of multiple CVDs other than MI. In this context, tako-tsubo cardiomyopathy (TC), which causes extensive acute regional wall motion dysfunction [69] and cannot be diagnosed based on ECG alone, results in a modest increase in cTn levels [69,70,71,72,73], with the peak occurring within 24 h and a more rapid decline in levels than seen in ACS. However, the elevation in levels of cTn do not reflect the dysfunction characteristic of TC [69]. Therefore, cTn release in TC appears to be unrelated to ischemic cardiomyocyte necrosis, which might explain why cTn have no prognostic value in TC [74,75]. In acute and chronic HF, the likely cause of troponin release is increased ventricular preload causing myocardial strain [76]. While the etiology is not specified through the rise in cTn [77,78,79], there is a strong prognostic value as the levels of increased cTn are closely linked with the severity of HF [80,81]. Furthermore, patients with aortic dissection display similar increases in cTn to that seen in MI associated with ACS, making cTn a poor diagnostic tool. Similarly, aortic stenosis and other valvular diseases display elevated TnT and TnI levels as reported in a significant body of literature [82]. For example, it was reported that hs-TnT plays a role as an independent prognostic factor for patients with aortic stenosis after valve replacement surgery [83]. Cardiac inflammatory conditions can also show increased cTn levels, most specifically in acute pericarditis through the involvement of the epicardium in the inflammatory process [84,85,86]. Furthermore, a stroke–both ischemic stroke and intracerebral haemorrhage–can cause an increase in cTn through secondary damage to the heart or a concomitant primary ACS [87,88,89]. Patients with acute pulmonary embolism (PE) also display increased cTn levels. Although the reason for this is not entirely understood, one explanation could be hypoxaemia resulted from perfusion–ventilation mismatch, hypoperfusion and paradoxical embolism from systemic veins to coronary arteries [90,91]. In patients with PE, cTnT peaks at lower levels and persists for a shorter period of time compared with cTnT values in acute MI [92]. Finally, elevated hs-cTnT precedes the advancement of hypertension and potentially identify people at future risk for hypertension and hypertensive end-organ damage [93,94,95,96,97]. However, there are no recommendations to measure hs-cTnT in screening for hypertension. Thus, more research is needed to establish the clinical use of troponins in the context of hypertension.

#### Disadvantages of Cardiac Troponins as a CVD Biomarker

Cardiac troponins are not specific to CVD as they can increase in various other conditions. These include sepsis, acute respiratory distress syndrome, chronic obstructive pulmonary disease, renal failure, diabetes, acute neurological event, chemotherapy, drugs, and toxins [60,98,99]. Additionally, cTn increase does not indicate a specific CVD etiology, e.g., whether it indicates MI, coronary artery disease, myocardial injury, or chronic dysfunction related myocardium or non-acute coronary syndrome ACS reasons. Therefore, the need for improved marker options yielding greater specificity remains obvious.

In addition, there are technical issues related to methods used to measure cTn. For instance, the use of high-sensitivity assays to measure troponins leads to the detection of cTn in the wider population with or without CVDs [100]. The efficiencies of cTn assays are also influenced by comorbidities, which cause measurement of inaccurate cTn circulation levels as seen in severe haemolysis [48,101,102,103]. Analytical issues related to heterophilic antibodies or interference of autoantibodies can cause false-positive or -negative results of cTn, which otherwise cannot be justified despite thorough clinical assessments [99,103,104]. There are many other analytical issues that interfere with the result and should be considered, such as non-specific binding, selection of matrix, and lot-to-lot variation. Thus, improving methods to measure troponins in order to avoid such issues is an important area for research.

### 3.2. Cardiac Natriuretic Peptides

Natriuretic peptides (NPs) are a group of hormones secreted from cardiomyocytes and are responsible for a spectrum of protective roles to the cardiovascular system including diuresis, natriuresis, and vasodilation, as well as opposition of the effects of the aldosterone-renin system [105,106]. Natriuretic peptides also influence metabolic processes such as lipolysis, weight loss, and improved insulin sensitivity [106]. Abnormalities of NPs are associated with multiple cardiovascular pathologies including HF, atrial fibrillation, systemic arterial hypertension, and inflammatory cardiac diseases, and therefore they are used as biomarkers to evaluate CVDs pathogenesis, diagnosis, prognosis, and therapy [107]. The main cardiac NPs in humans include atrial natriuretic peptide (ANP), B- type natriuretic peptide (BNP), and C- type natriuretic peptide (CNP) [108], which constitute a well-integrated regulatory system, and possess similar structural conformation but different potency [108,109].

The transcriptional regulation of ANP and BNP in humans is controlled by the heart cells [110,111]. C-type natriuretic peptide transcription is not controlled by any specific type of cardiomyocytes as it is derived primarily from endothelial cells and it presents within atrial and ventricular myocardium [112]. In pathological conditions, the key stimulant for a release of ANP and BNP from the heart is myocardial mechanical stretch and neurohormonal activation. Both BNP and ANP are elevated as a response to ventricle wall stress induced by stretched myocytes because of pressure overload or volume expansion of the ventricle, which are present in conditions of HF and MI [113,114,115]. The release of BNP or ANP leads to improved myocardial relaxation by inhibiting the vasoconstriction, fluid retention, and antidiuretic actions of the stimulated renin-angiotensin-aldosterone system [105,106,114,116].

All natriuretic peptides are synthesized as preprohormones, which are cleaved into an inactive N-terminal fragment (NTproANP, NT-proBNP) and the biologically active ANP [109,117,118,119,120] or BNP [116,121,122]. Both NT-proBNP and NT-proANP as well as biologically active BNP and ANP could be found in plasma [105,123,124]. The active ANP has a very short half-life of less than 5 min [125,126,127], which makes the use of NT-proANP more reliable as a biomarker because it is secreted correlatedly with ANP but has a markedly longer half-life of 60–120 min [117,128]. However, because NTproANP can be further cleaved into smaller fragments in vivo, the determination of mid-regional proANP (amino acids 53–90; MR-proANP) is the favorable detection site of ANP [117,129,130,131]. The half-life of BNP and NT-proBNP is 20 min and 90–120 min, respectively [124,132,133,134]. Circulating levels of BNP and NTproBNP can be used in patient care as they are correlated.

Elevation circulating levels of NPs’ levels helps in the detection of CVDs and is associated with worse prognosis regardless of the etiology of the increase [105]. In fact, NPs are responsible for the classification of the severity, influencing care plans, and evaluating the prognosis of heart disease [132,135,136,137,138,139,140,141]. Both BNP and NT-proANP levels are reported to be independently predictive of the risk of death from major cardiovascular event, HF, stroke or transient ischemic attack, and atrial fibrillation [142]. Additionally, in apparently healthy individuals, NTproBNP gives prognostic information of mortality and first major cardiovascular events that goes further than classical risk factors [143].

Natriuretic peptides are increased to very high blood levels in acute HF and to a lesser extent in chronic HF. For diagnosing acute and chronic HF, according to the guidelines of the ACCF/AHA/ESC, BNP and NT-proBNP are considered to be the most valuable and reliable biomarkers, with proportional increase to the severity of the disease giving information about HF prognosis [141,144,145,146]. This increase is because patients might actually manifest a state of BNP insufficiency, attributed to both a deficiency of active BNP and resistance to its actions [105,108]. However, ANP is also secreted in patients with chronic HF; its release is less marked than that of BNP and is suppressed in severe cases by BNP increase [144,147,148]. On the other hand, MR-proANP measurement can provide valuable additive information in HF assessment and it correlates with ventricular wall stress and HF severity [149,150].

As NPs affect multiple aspects of the cardiovascular system, their levels increase in various CVDs. Among these CVDs, acute ischemic heart diseases including MI are associated with an elevation of BNP and NT-proBNP levels, which is of value in predicting the prognosis and severity [105,151,152]. After a MI, NT-proBNP is increased for about 12 weeks and therefore might be more useful than BNP as a diagnostic and prognostic biomarker [113,153]. Furthermore, BNP and NT-proBNP are related to arrhythmias as both are increased in atrial fibrillation patients [154,155,156]. Parallel rise in the circulating levels of NPs and troponins were also reported in patients with various tachycardias [157]. Evidence from animal experiments suggests that BNP mRNA and its protein increase as early as 10 min after transient lethal ventricular arrhythmias, possibly mediated by myocardial stretch [158]. In addition, the levels of BNP are considerably elevated in TC, with a value of early BNP/cTnT and BNP/CK-MB ratios to distinguish TC from acute MI with higher accuracy than BNP alone [71]. This highlights that combining the determination of BNP levels with other biomarkers could provide additional benefit for the differential diagnosis of certain heart diseases. Patients with valvular diseases including aortic stenosis show increased NPs levels which are useful for clinically monitoring patients’ pre- and post-valvular surgery. In general, levels of NPs are likely to rise with augmenting severity of the valvular dysfunction and with the consequential cardiac remodeling, giving significant prognostic details that could guide risk stratification and scheduling for surgery [159,160,161,162,163,164,165]. Elevated levels of NPs can also be seen in the pulmonary embolism, likely reflecting a right heart strain [166], although not usually to the same extent as in HF [166,167]. Additionally, patients with other pulmonary diseases can have elevated BNP levels but have much lower levels than in patients with HF [168,169,170,171,172]. Indeed, right heart abnormalities resulted from severe lung illness can lead to increased circulating levels of NPs [169,171]. Finally, the levels of NPs potentially present important information about the severity of hypertension [173]. One study reported a prognostic value for NT-proBNP to detect underlying cardiac structural damage in asymptomatic hypertensive patients [174], whilst another study suggested that BNP levels can aid in predicting the presence of ventricular wall dysfunction [126].

#### Disadvantages of Natriuretic Peptides as CVD Biomarkers

Increased circulating concentrations of NPs in circulation for CVD diagnosis are not adequately accurate or specific. In fact, expression of NPs is not exclusive to the heart. ANP is detected in the lungs, brain, liver, and in several cell types such as immune cells [175,176]. Similarly, BNP in humans is expressed in the brain, lungs, kidneys, aorta, and adrenal glands [176,177]. However, outside the heart, both ANP and BNP concentrations are much smaller. Overall, their diagnostic value is often complicated by increased levels due to non-cardiac conditions–some of which are potentially life-threatening. Such examples of non-cardiac pathologies include acute or chronic renal failure [178,179], ascitic liver cirrhosis [180,181], respiratory diseases [182], obesity, and endocrine disorders such as hyperaldosteronism [183,184], adrenal tumours [185], and hyperthyroidism [186,187]. Furthermore, a number of physiological elements are associated with higher levels of NPs, including increased age and female gender [188,189,190,191].

## 4. Future: Prospective Biomarkers for CVDs

### 4.1. Heart-Type Fatty Acid-Binding Protein (H-FABP)

Heart-type fatty acid-binding protein (H-FABP) belongs to the FABP family and plays a role in multiple cellular activities including cardiac lipid transport, regulation of cell proliferation, and myocyte homeostasis [192,193,194,195,196,197]. The FABP family is detected in multiple tissues that demand increased activity of fatty-acids; H-FABP is mostly translated in the heart [198]. High amounts of H-FABP are found in muscle cells and are secreted very quickly after heart damage because of its low molecular size and free cytoplasmic location [199,200,201]. No cross-reaction was observed between H-FABP and other FABP types because of H-FABP’s differences in morphology and immunology. Consequently, H-FABP displays a significant specificity for myocardial damage diagnosis [202].

Indeed, several reports have highlighted H-FABP as a specific former marker of myocardial damage. Compared to cardiac troponins, H-FABP circulation levels are elevated within three hours of the onset of acute MI, and reciprocate within 20 h [200,203,204,205]. Additionally, circulating H-FABP is significantly increased after 15 min of induced MI by transcoronary ablation of septal hypertrophy in patients with hypertrophic obstructive cardiomyopathy [199]. However, mixed data are available about its added value in MI. Some studies suggest incremental value of using H-FABP in conjunction with hs-troponins [206,207], while other studies do not indicate additional advantages of H-FABP levels with cTn for diagnosing acute MI [208,209,210].

Various reports have evaluated the function of H-FABP in patients with HF, suggesting an impartial association between H-FABP and HF outcomes, as well as the risk of detrimental cardiac injuries [141,211,212,213,214,215]. Higher levels of H-FABP are detected in patients with arrhythmia and HF [216]. Similarly, an impartial correlation of H-FABP with clinical outcomes has been documented in hypertensive patients with aortic valve disease [200]. Furthermore, cardiac remodeling characterised by ECG is found to be correlated with levels of H-FABP in patients that exhibited troponin levels below the current cut-offs [200]. It is worth mentioning that valvular replacement in patients with severe aortic valve stenosis is suggested to significantly reduce H-FABP plasma concentration, pointing at decreased ventricular wall stress and the prospective to ameliorate cardiac remodeling [217]. Moreover, H-FABP plays a role in risk assessment of PE because it indicates right-ventricular strain early [91,218,219,220]. Additionally, significant correlation between H-FABP and the risk of major adverse events and mortality was showed in PE patients, which explains why the 2019 ESC guidelines on the diagnosis and management of acute PE recommend the use of H-FABP for prognosis assessment [221], despite the fact that prospective trials are yet to be initiated.

Disadvantages of H-FABP as a CVD biomarker include unclear results in renal dysfunction caused by the fact that H-FABP elimination is completely dependent on the kidneys. In addition, H-FABP has low specificity in the presence of skeletal muscle injury as it increases in skeletal muscles pathologies [17].

Ultimately, no agreement for measuring H-FABP levels has been agreed upon in routine clinical practice in the context of CVDs. Future applications of H-FABP could involve initial investigation of ischemia and guidance of long-term treatment planning. In addition, H-FABP may play an essential role in diagnosis of HF, combined with clinical assessments, imaging, and a multi-biomarker approach.

### 4.2. Copeptin

Copeptin, a neurohormonal marker of stress, is the 39-amino acid C-terminal portion of arginine vasopressin (AVP) precursor peptide that is released from the posterior pituitary gland into the bloodstream upon appropriate stimuli [222]. The rate of release of copeptin is directly linked to the release of AVP, which is crucial to preserve the water balance of the body. Indeed, AVP–also known as antidiuretic hormone (ADH)–controls the kidney reabsorption of free water, total volume of blood, osmolality of body fluid, vasoconstriction, and myocardial contraction, as well as cell proliferation [223]. AVP has a low molecular weight of nine amino acids and a very short half-life of 24 min [224], therefore its levels are better represented by circulating copeptin, which is more reliable to determine [225]. Indeed, copeptin has days of stability after blood withdrawal and is measured by a sandwich immunoluminometric assay within 3 h [225].

Copeptin is secreted at the onset of MI [226,227,228]. However, there are controversial data related to the usefulness of copeptin in MI diagnosis, with some data suggesting its value in identifying low-risk patients presenting to the emergency department with a potential acute MI [227,229]. Contrarily, other studies point out that if the rapid rule-out hs-cTn assays are available, copeptin has limited added value [72,230]. Further trials are needed to explore the potential benefits of the combination of copeptin with hs-cTn. Moreover, copeptin levels may provide prognostic information in combination with cTn [231,232]. Copeptin levels are also increased in several CVDs, including the development of HF where levels of circulating AVP are increased [233,234,235,236]. In fact, copeptin has been demonstrated to be a valuable prognosticator of the outcome and severity of HF, as well as independent prognosticator of mortality, generally indicating a poor prognosis [231,237,238,239,240,241,242,243]. Increased levels of copeptin are also seen in PE [91] and acute aortic syndrome [244], as well as hypertension [245,246,247].

Overall, copeptin is not a specific marker for CVD and it appears to be a complementary biomarker, which could be used as part of a multi-marker risk prognosis panel to better guide clinical decisions. However, more research is needed to confirm its application in routine practice.

Disadvantages of copeptin as a CVD biomarker include the large number of other disorders with which its elevated levels are associated, including respiratory disorders (e.g., acute exacerbation of chronic obstructive pulmonary disease, lower respiratory tract infections, and acute dyspnea), sepsis, hemorrhagic and septic shock, diabetes, metabolic syndrome, hyponatremia, vasodilatory shock, diabetes insipidus, polycystic kidney disease, intracerebral hemorrhage, ischemic stroke, and traumatic brain injury [223,248].

### 4.3. Adrenomedullin (ADM)

Adrenomedullin (ADM) is a peptide synthesised in many tissues–especially vasculature structural cells–and has numerous biologic actions, including being a renal vasodilator with natriuretic and diuretic properties [249,250,251,252,253,254]. ADM is derived from a precursor peptide (proADM) by post-translational processing, giving two cleaved inactive fragments (the mid-regional part of proADM (MR-proADM) and the C-terminus of the molecule) [255]. Titration of ADM secretion in the circulation is challenging because of the instant coupling of ADM to receptors in the proximity of its generation, inducing autocrine or paracrine activities. Additionally, its circulating levels are reduced by the short half- life of ADM and its instability [255,256]. Therefore, it is more practical to measure the levels of MR-proADM, which is more stable and directly reflects the ADM levels [255,257,258].

In acute MI, ADM levels were reported to increase in the early phase reaching a maximum after 2–3 days and returning to baseline after three weeks [259]. Furthermore, ADM is increased in HF [260,261,262], related to the severity and prognosis of HF, as well as to ischemic HF with ventricular dysfunction [251,258,261,262,263,264,265,266,267,268].

ADM is also far from being specific to CVDs since it is increased in various other diseases such as renal dysfunction, diabetes, and septic, haemorrhagic, or cardiogenic shock [269]. However, in patients with hypertension, ADM elevation directly correlates with the stages of hypertension as defined by the World Health Organization (WHO), as well as with the severity of target organ damage. This suggests a potential value for measuring ADM levels in hypertension, although the detailed relationship between plasma ADM levels and blood pressure is yet to be understood.

### 4.4. P-Selectin

P-selectin is a cell-surface adhesion molecule and is expressed on activated endothelial cells and platelets after stimulation by several inflammatory stimuli [270]. P-selectin has been suggested to participate in the pathology of atherosclerosis, a key risk factor for CVDs [271,272,273,274]. In fact, in patients with coronary artery disease or after acute MI, elevated levels of P-selectin are reported and hold an association with poor prognosis [275,276,277,278]. Furthermore, P-selectin was shown to provide incremental diagnostic value when used in combination with troponin and ECG to exclude acute MI with relatively high sensitivity [275]. Interestingly, increased circulating P-selectin levels are also linked to a higher risk of developing acute MI, stroke, and CVD-related death in healthy women [279]. Of note, P-selectin is also found to increase with exercise and age [280]. In this context, the value of P-selectin as a CVD biomarker remains to be clarified, independently of other established biomarkers that have demonstrated value for both screening and prognosis of atherosclerosis, i.e., cardiac troponins, natriuretic peptides, or C-reactive protein (CRP).

### 4.5. Soluble Urokinase -Type Plasminogen Activator Receptor (suPAR) and Plasminogen Activator Inhibitor-1 (PAI-1)

Soluble urokinase plasminogen activator receptor (suPAR) is a proinflammatory biomarker related to immune response and fibrinolysis inhibition, implicated in atherosclerosis [281,282,283,284]. Moreover, plasminogen activator inhibitor (PAI-1)–a potent inhibitor of fibrinolysis–is also linked with atherosclerosis, excessive fibrin accumulation and thrombus formation.

Increased circulating suPAR and PAI-1 levels are correlated with increased risk of major adverse CVD events, including death and MI in advanced coronary artery disease, as well as the early phase of atherosclerosis [285,286,287,288]. Indeed, in patients with MI, increased suPAR levels independently predict all-cause mortality and recurrent MI [289]. Similarly, increased PAI-1 concentrations were an independent risk factor for major CVD events after successful coronary stenting [290]. Moreover, elevated suPAR levels are linked to increased CVD risk in apparently healthy individuals [291,292]. However, association between suPAR and disease development and mortality in the general population was not specific to CVDs, since circulating levels of suPAR and PAI-1 were also associated with the development of inflammatory diseases, cancer, type 2 diabetes, and mortality in the general population [291,293].

### 4.6. Extracellular Matrix Remodelling

#### 4.6.1. Galectin-3 (GAL-3)

Galectin-3 (GAL-3) is a member of the β-galactoside-binding lectins family, which is implicated in altering “cell-to-cell” and “cell-to-matrix” interactions, as well as regulating cellular functions at the cell surface [294,295,296,297]. GAL-3 is also involved in cell adhesion, proliferation, apoptosis, and angiogenesis. Notably, GAL-3 enhances fibrosis and its circulation levels are associated with cardiac remodelling and ventricular hypertrophy [298]. In fact, elevated GAL-3 levels aid in detecting early stage of myocardial dysfunction and HF, as well as other heart diseases, including ACS and acute myocarditis [299,300,301,302,303,304,305,306,307,308,309,310,311,312,313,314,315]. Moreover, this increase in GAL-3 levels is linked to a worse prognosis in these conditions [316,317], representing a potential sensitive biomarker for CVDs [301,302,313,314,315]. As a novel biomarker, GAL-3 has been recommended by the ACC/AHA guidelines for assessment of myocardial fibrosis in HF, whereas the ESC has not recommended the clinical use of GAL-3 [146]. Left ventricular remodelling following acute MI was found to hold no correlation with serum GAL-3 levels [302]. In addition, GAL-3 could not predict the mortality in patients with HF compared to other biomarkers, while it showed a distinctive advantage when used as in a member of multi-biomarker panel [318].

As with other potential CVD biomarkers, GAL-3 is also detected in many non-cardiac disorders, including kidney disease [319,320,321,322], diabetes [319,320,323], viral infections [306,324,325,326], autoimmune diseases [327,328,329,330], neurodegenerative disorders [331,332,333,334,335], and tumour formation [336,337,338,339,340,341,342,343,344,345]. Collectively, the usefulness of GAL-3 to diagnose cardiac disorders is limited, and more research is needed to confirm its advantages as a prognostic CVD biomarker.

#### 4.6.2. Matrix Metalloproteinases (MMPs) and Their Tissue Inhibitors (TIMPs)

Matrix metalloproteinases (MMPs) are enzymes that degenerate the formational constituents of the extracellular matrix and divert biologically active elements, such as growth factors, cytokines, and chemokines [346]. The actions of MMPs are closely regulated by the endogenous tissue inhibitors of metalloproteinases (TIMPs). The interaction’s equilibrium between MMPs and TIMPs is responsible for maintaining tissue homeostasis and changes in this equilibrium are involved in progression of CVDs [347,348,349,350,351,352,353,354]. Furthermore, members of MMPs and TIMPs are proposed as promising biomarkers to predict future CVD events. For instance, in patients with coronary artery stenosis, serum levels of TIMP-1 were reported to exhibit a correlation with the incidence of major adverse cardiac events, and, thus, the prognosis in these patients [355,356,357]. Additionally, circulating levels of TIMP-1 rise in males more than in females, and elevate with well-known CVD risk factors, such as age, body mass index, the ratio of lipoprotein cholesterol, smoking, and diabetes [358]. Another example is MMP-3, which was shown to be associated with MI in one study [302]. Similarly, MMP-8 levels were associated with the risk for a coronary artery disease event, MI and death [359]. Of note, MMP-9 represents the most studied MMP in relationship to CVDs, with its levels reflecting the severity of the infarction and predicting mortality in MI patients [360,361,362], as well as the progression of HF [363,364]. Moreover, in hypertension, induced-MMP-9 at a very early stage enhances collagen breakdown and arterial destruction, with hypertensive patients presenting increased serum MMP-9 levels, which were related to aortic stiffness [365]. Further investigations are necessary to determine the advantages of the use of specific MMPs and TIMPs for diagnosis and prognosis of CVD in clinical practice.

### 4.7. Inflammatory Markers

#### 4.7.1. Growth Differentiation Factor 15 (GDF-15)

Growth Differentiation Factor 15 (GDF-15) is a stress-responsive cytokine of the transforming growth factor -β (TGF- β) superfamily, constituting a biomarker of inflammation, as well as oxidative stress and hypoxia. Of note, CVDs are a main reason for GDF-15 upregulation, which is hypothesized to reflect a defense mechanism in both acute and chronic cardiac injury [366,367]. Interestingly, the biological actions of GDF-15 are dependent on the underlying pathophysiology and may differ with the stage of the illness [366,368,369,370,371,372]. In fact, increased serum GDF-15 levels are observed in acute MI [366,373] and acute HF [374,375]. Furthermore, GDF-15 levels have been associated with adverse events in community-dwelling populations and higher risk to develop CVDs in patients with non-CVDs disorders [376,377,378,379,380,381,382,383,384,385], as well as adverse prognosis in patients with ACS [386,387,388,389] and HF [63,318,375,390,391,392]. Moreover, serial measurements of GDF-15 have been found to improve risk prediction models in patients with HF [390]. In addition, in patients with atrial fibrillation, increased GDF-15 levels are associated with major bleeding, mortality, and stroke [154,393,394,395,396]. Accordingly, GDF-15 has a potential valuable role in assessing the risk of major bleeding in patients treated with oral anticoagulant therapy [397]. Collectively, GDF-15 could be utilized for screening and may have prognostic value, especially as in the context of a composite biomarker panel [7,378,380,385,386,390,391,396,398,399,400]. Further research is necessary to evaluate the role of GDF-15 for guiding clinical management decisions and treatment options for CVD, in comparison to and in combination with other CVD biomarkers and clinical predictors.

However, GDF-15 expression is not specific to the myocardium, since this cytokine is also expressed in various other tissues/cells such as macrophages, endothelium, smooth muscle, and adipocytes. Thus, levels of GDF-15 may also rise in different conditions that are not related to CVDs, such as advanced age, obesity, diabetes, and kidney dysfunction [401,402,403,404,405], limiting its diagnostic value for CVDs [404,405].

#### 4.7.2. Endothelin-1 (ET-1)

ET-1 is a potent hormone with multiple actions regulating vasoconstriction and renal sodium excretion [406,407,408,409], and is produced by many organs, including the heart and kidneys. Of note, the major source of ET-1 is the vascular endothelium, with ET-1 being constantly synthesised in the blood vessels to preserve vascular resistance and pressure, as well as to regulate blood pressure [407,410]. As such, ET-1 levels have been shown to correlate with inflammation, vasoconstriction, vascular and cardiac hypertrophy, and with the development and progression of CVDs [406,411,412]. In addition, both ET-1 and C-terminal proET-1 levels have been shown to increase with age [413,414,415,416,417], lung function [418,419,420], chronic kidney disease [413,415,416,421], and cancer [422]. Moreover, elevated ET-1 levels are associated with HF, coronary artery disease, hypertrophic cardiomyopathy, hypertension, and cervical artery dissection [307,411,423,424]. Indeed, ET-1 has been demonstrated to have a potential use as a prognostic biomarker in many of these CVDs, including mortality in patients following acute MI and diagnosis of congestive HF. For example, in patients with HF, measuring ET-1 levels added prognostic information that was supplementary to that provided by NT-proBNP [425]. Interestingly, the ET-1 active form was initially thought to be unstable in serum with a short plasma half-life of 1.4–3.6 min [426,427], which restricts its clinical use. Later on, ET-1 was reported to have a large volume of wide distribution and an extended half-life of 7.5 h [428]. Therefore, the developed C-terminal segment of pro-Endothelin-1 (CTproET1) and other ET-1 surrogate peptides as the more stable form of ET-1 may not be as disadvantageous as has been initially suggested. Notably, in patients with chronic coronary artery disorder or acute MI, CT-proET-1 has been linked with cardiovascular death and HF independent of clinical variables, and demonstrated prognostic value comparable to BNP or NT-proBNP [267,429,430]. Additionally, increased blood levels of C-terminal proET-1 and ET-1 have been correlated with larger left atrial size, and all-cause mortality in samples of general populations [413,414], which potentially hold screening value. This, however, requires further study in diverse populations. Hence, the use of ET-1 levels to select patients for primary preventive strategies and to guide personalised treatment regimens are future directions to be explored.

#### 4.7.3. Suppression of Tumorigenicity 2 (ST2)

Suppression of Tumorigenicity 2 (ST2) is a cellular receptor for interleukin-33 (IL-33) and belongs to the interleukin-1 receptor family [431]. ST2 has two variants, including the transmembrane (ST2L) and soluble (sST2) isoforms, which can be detected in circulation [432]. Notably, ST2 is transcripted by cardiomyocytes and vascular endothelial cells along with its agonist IL-33 following myocardial damage [431,433]. Coupling of IL-33 and ST2L potentially suppresses hypertrophy, fibrosis, and apoptosis of the heart, and alleviates detrimental cardiac remodelling [431,433]. However, sST2 competes with ST2L to bind IL-33, leading to suppression of the protective effects of the ST2L-IL33 pathway to the heart [431,433]. In fact, it was reported that blood levels of sST2 are markedly elevated in HF onset and worsening of chronic HF–as well as with MI–aortic valve impairments and hypertension [217,401,434,435,436,437,438,439]. Therefore, sST2 represents a potential predictor of CVDs prognosis [217,401,434,435,436,439]. Indeed, measuring sST2 is recommended as prognostic marker in HF according to ACCF/AHA guidelines [146,440].

Although sST2 can play an important role as a prognostic marker in the context of CVDs, it has limited diagnostic value because increased sST2 levels are not specific for any human disease. Indeed, such high sST2 levels may occur not only in CVD, but also in pulmonary diseases, burn injuries, and immune diseases [432]. Additionally, the exact origin of circulating sST2 is not clear, and studies did not prove that cardiac cells were solely responsible for serum sST2 changes in CVDs [441,442,443,444]. Indeed, sST2 is expressed by several tissues, such as the colon and many haematopoietic cells (e.g., basophiles, CD4 lymphocytes, eosinophils, macrophages, and T-helper 2 cells) [432]. Overall, the contribution of non-cardiac secretion to the total circulating sST2 levels, as well as the pathophysiologic implications of sST2 in CVDs, is not well studied yet and represents an area for further research.

#### 4.7.4. Lipoprotein-Associated Phospholipase A2 (Lp-PLA2)

Lipoprotein-associated phospholipase A2 (Lp-PLA2) is secreted by inflammatory cells that are involved in forming vulnerable plaques and developing atherosclerosis [445,446]. In fact, elevated Lp-PLA2 levels are associated with increased risk of coronary events [447], unrelated to the levels of total cholesterol [448]. In patients diagnosed with CVDs, levels of Lp-PLA2 also predict adverse outcomes indicating its prognostic value [449]. Further investigations are needed to confirm the potential value of Lp-PLA2 as a biomarker for CVDs.

#### 4.7.5. Soluble CD40 Ligand

Soluble CD40 Ligand (sCD40L) is a mediator of vascular inflammation that is implicated in atherogenesis, activating CD40 receptor on various cells that contribute to atherosclerosis progression (e.g., on macrophages, endothelial cells and T-cells) [450,451]. Notably, CD40L antibodies and CD40L deficiency have been associated with reduced atherosclerosis [452]. Furthermore, sCD40L concentrations have been associated with prognosis in ACS [453,454]. However, the current knowledge about sCD40L is limited, and exploring its value for the clinical practice related to CVDs is needed.

### 4.8. MicroRNAs

Out of the three billion base pairs in the human genome, only about 1% directly code for proteins. The remainder were previously thought to be ‘junk’ DNA, but have now been recognised to play important roles in gene regulation and function. Some of these non-coding RNAs include short microRNAs (miRNAs) and longer, long non-coding RNAs (IncRNAs).

miRNAs are approximately 22 nucleotides in length and were first discovered in 1993. Since then, the field has expanded exponentially with 2000 miRNAs having been identified in humans (http://www.mirbase.org (accessed on 14 April 2022). A key feature of the mechanism of action of miRNAs is that a single miRNA can regulate the expression of several genes, whilst, conversely, individual genes can be regulated by different miRNAs [455]. The factors affecting this regulatory mechanism are highly complex and are still not well-understood. Nevertheless, multiple basic science studies have found that miRNAs play a role in normal cardiac development, as well in CVDs, including (with key implicated miRNAs in parenthesis) HF (miR-133, miR-1, miR-25), atherosclerosis (close to 70 different miRNAs), arrhythmias (miR-1, miR-328, miR-223, miR-664), and hypertension (miR-181a, miR-663, miR-132, miR-212, miR-143/145)

Notably circulating miRNAs are highly stable and have been reported to be implicated in multiple cardiovascular conditions. In terms of clinical studies, the expression of each one of miR-21, miR-486-5p, miR-146a, miR-664a-3p, miR-195, miR-217, miR-126, miR-143, miR-146a, and miRNA-210 was altered in patients with severe vascular disease and AMI, as well as atherosclerosis. For example, in HF, altered circulating levels of miRNAs were reported for miR-122, miR-210, miR-423-5p, miR-499, miR-622, miR-16, miR-27a, miR-101, and miR-150. Such miRNAs are suggested to play a role as diagnostic and prognostic CVD biomarkers [456,457,458,459].

Although recent discoveries in miRNA research highlight their diagnostic, prognostic and perhaps even therapeutic value in CVDs, there is still lack of understanding regarding how exactly these miRNAs regulate gene expression. Additionally, the variability in miRNA-based phenotype regulation in CVD, the interactions between multiple miRNAs with their shared cognate mRNAs, and the effect of comorbidities on the circulating miRNAs levels are still important areas that need further investigations [456,457,458,459].

### 4.9. Other Biomarkers

Finally, traditional inflammatory biomarkers are involved in the pathology of CVDs, including serum amyloid A [460], osteoprotegerin (OPG) [307,434,461,462], myeloperoxidase, CRP, erythrocyte sedimentation rate (ESR), cytokines, neutrophils, and monocytes [463]. Furthermore, well-described biomarkers of diseases other than CVDs provide important information about the progression of a given CVD. For instance, since there is a close relationship between functional regulation of the kidneys and the heart, it is not surprising that markers related to kidney function such as cystatin C, uric acid, and albuminuria play a role in evaluating patients with various CVDs (e.g., in patients with HF). Similarly, markers of metabolic disorders are considered to have a prognostic information for the severity of CVDs, such as the lipid profile, vitamin D, fetuin-A, and diabetes-related biomarkers of, for example, blood glucose and haemoglobinA1c levels. These may be valuable additions in multi-marker approaches for effective screening, diagnosis, and prognosis of CVDs.

## 5. Conclusions

In conclusion, the contribution of measuring CVD biomarkers has become increasingly essential for the clinical practice, with an increasing number of established and emerging CVD biomarkers (Table 1). Such biomarkers are increasingly becoming more specific to cardiovascular pathophysiology, with enhanced sensitivity and specificity. For example, the National Institute for Health and Care Excellence (NICE) recommends measuring NT-proBNP in people with suspected HF to direct care plan. Urgent referral to have specialist assessment and transthoracic echocardiography is suggested when NT-proBNP level is above 2000 ng/L [464]. Additionally, according to the 2017 ACC/AHA/HFSA guidelines for the use of biomarkers in the management of HF, measurement of BNP or NT-proBNP was recommended to diagnose HF class I according to the New York Heart Association (NYHA) functional classification, as well as at hospital admission and discharge for added risk stratification in HF class I and IIa, respectively [465]. Furthermore, Tns are recommended to evaluate hospital admission prognosis in HF class I, whilst sST2 and GAL-3 received a class II recommendation for risk prediction in HF [465]. In addition, other novel biomarkers, such as MR-proANP and ADM, have been reported to be independent predictors of HF and to correctly reclassify HF patients [241,264]. Currently, the role of established CVD biomarkers has progressed from the simple retrospective confirmation of an already diagnosed condition to a central position in the screening/diagnostic/prognostic clinical algorithms. However, the use of many novel CVD biomarkers has yet to be employed in the clinical practice. This partly due to still insufficient knowledge regarding their exact clinical usefulness and their potential advantages over existing established biomarkers. Moreover, there are also technical issues (e.g., lack of readily commercial clinical assays) and challenges regarding the precise determination of any novel biomarker as an analyte. Therefore, intensive research is still required in this field to address the existing gaps in our knowledge regarding the clinical use of the aforementioned novel biomarkers in the context of CVD. Indeed, clear evidence is needed to prove the potential benefit of such novel biomarkers for CVD screening/diagnostics/prognostics, as well as potential benefits for personalized/biomarker-guided therapies. In this setting, multi-biomarker approaches, employing a patho-biologically diverse set of biomarkers, could have a significant impact regarding the diagnosis and/or management of CVDs. 

## Figures and Tables

**Table 1 ijms-23-05680-t001:** Details of selected key established and emerging biomarkers for cardiovascular diseases (CVD). For myocardial infarction (MI), troponins (Tns) are the main biomarker used in current clinical practice for diagnosis, along with natriuretic peptides (NPs) for prognosis. Candidate biomarkers that provide additional information for prognosis in MI include Adrenomedullin (ADM), Heart-Type Fatty Acid-Binding Protein (H-FABP), Copeptin, P-selectin, Soluble Urokinase -type Plasminogen Activator Receptor (suPAR), Plasminogen Activator Inhibitor-1 (PAI-1), Galectin-3 (GAL-3), Matrix metalloproteinases (MMPs) and their tissue inhibitors (TIMPs), suppression of tumorigenicity 2 (ST2), Growth Differentiation Factor 15 (GDF-15), and Endothelin-1 (ET-1). Indeed, diagnosis and prognosis of coronary syndromes are currently guided by Tns, NPs, whilst P-selectin, suPAR, PAI-1, and GAL-3 could be additionally useful in the context of acute coronary syndromes in the future. Moreover, NPs levels are also used for diagnosis and prognosis of heart failure (HF), whilst Tns, FABP, Copeptin, ADM, MMPs, TIMPs, ST2, GDF-15, ET-1, and GAL-3 could also be helpful as biomarkers for HF screening and prognosis. Similarly, for diagnosis and prognosis of arrhythmias, specifically atrial fibrillation, a number of biomarkers, including Tns, NPs, H-FABP, GDF-15, and pro-inflammatory cytokines are described below. Regarding pro-inflammatory cytokines, interleukin-6 (IL-6), interleukin-8 (IL-8), tumor necrosis factor-α, and intercellular adhesion molecule-1 impact arrhythmogenic activity and could be prognosis biomarkers [466,467,468,469]. Indeed, increased levels of IL-6 and interleukin-10 were predictive of the risk of atrial and ventricular arrhythmias in patients hospitalized with COVID-19 [470]. Interestingly, NPs levels could be also assessed for prognosis of heart surgical procedures or congenital heart disease. Furthermore, tako-tsubo cardiomyopathy or acute pericarditis diagnosis can be aided by Tns levels, whilst measuring Tns, NPs, copeptin, H-FABP, MMPs, TIMPs, and ST2 may be helpful for the diagnosis and prognosis of aortic dissection, acute aortic syndrome, aortic stenosis, and other valvular diseases. Finally, circulating levels of Tns, NPs, Copeptin, and P-selectin are utilized for the diagnosis of stroke, while Tns H-and FABP are used for pulmonary embolism diagnosis and prognosis, respectively.

CVD Biomarker	Use	Present	Future	S	D	P	Ref.
Troponins (Tns)	Myocardial infarction	**✓**			**✓**	**✓**	[14,51,471,472]
	Coronary syndromes	**✓**			**✓**	**✓**	[473,474,475,476,477]
	Heart failure	**✓(?)**		**✓**		**✓**	[80,81,478,479,480,481,482,483]
	Atrial fibrillation	**✓(?)**			**✓**	**✓**	[484,485,486,487]
	Tako-tsubo cardiomyopathy	**✓(?)**			**✓**		[69,70,71,72,73,74,75]
	Aortic dissection, Aortic stenosis and other valvular diseases	**✓(?)**			**✓**	**✓**	[82,83]
	Acute pericarditis	**✓(?)**			**✓**		[84,85,86]
	Stroke	**✓(?)**			**✓**	**✓**	[486,487,488]
	Pulmonary embolism	**✓(?)**			**✓**		[90,91,92]
Natriuretic Peptides (NPs)	Heart failure	**✓**		**✓**	**✓**	**✓**	[142,258,264,489,490,491,492,493,494,495]
	Coronary syndromes	**✓**				**✓**	[267,491,496,497]
	Myocardial infarction	**✓**				**✓**	[263,473,498,499,500]
	Atrial fibrillation		**✓**		**✓**	**✓**	[142,501,502]
	Stroke		**✓**		**✓**		[142,503]
	Surgical procedures involving the heart		**✓**			**✓**	[145]
	Pulmonary embolism		**✓**			**✓**	[166,167]
	Left ventricular hypertrophy		**✓**		**✓**	**✓**	[126,182,383]
	Valvular heart disease		**✓**			**✓**	[142,146,160]
	Congenital heart disease		**✓**			**✓**	[122,146,182]
Heart-Type Fatty Acid-Binding Protein (H-FABP)	Myocardial infarction		**✓**		**✓(?)**	**✓**	[504,505,506,507]
	Heart failure		**✓**			**✓**	[213,214,508]
	Arrhythmia		**✓**			**✓**	[211,216]
	Valvular heart disease		**✓**		**✓**		[200]
	Pulmonary embolism		**✓**			**✓**	[91,218,219,220]
Copeptin	Myocardial infarction		**✓**		**✓(?)**	**✓**	[226,227,232,509]
	Heart failure		**✓**			**✓**	[238,239,510]
	Stroke		**✓**		**✓**	**✓**	[511,512,513]
	Pulmonary embolism		**✓**	**-**	**-**	**-**	[91]
	Acute aortic syndrome		**✓**			**✓**	[244]
Adrenomedullin (ADM)	Myocardial infarction		**✓**		**✓**	**✓**	[266,268,514]
	Heart failure		**✓**			**✓**	[258,264]
Ischemia Modified Albumin (IMA)	Unstable angina		**✓**		**✓**		[515,516,517,518]
P-selectin	Myocardial infarction		**✓**	**✓**		**✓**	[275,276,277,278,279]
	Acute coronary syndrome		**✓**	**✓**		**✓**	[275,276,278]
	Stroke		**✓**	**✓**			[279]
Soluble Urokinase -type Plasminogen Activator Receptor (suPAR) and Plasminogen Activator Inhibitor-1 (PAI-1)	Myocardial infarction		**✓**			**✓**	[285,286,287,288]
	Acute coronary syndrome		**✓**			**✓**	[285,286,287,288]
Galectin-3 (GAL-3)	Myocardial infarction		**✓**			**✓**	[302]
	Heart failure		**✓**			**✓**	[313,314,519,520]
	Acute coronary syndrome		**✓**			**✓**	[301,302,313,314,315]
	acute myocarditis		**✓**			**✓**	[306,521]
Matrix metalloproteinases (MMPs) and their tissue inhibitors (TIMPs)	Myocardial infarction		**✓**	**✓**		**✓**	[364,522]
	Coronary artery stenosis		**✓**	**✓**		**✓**	[361,523]
	Heart failure		**✓**	**✓**		**✓**	[363]
	Multiple CVDs		**✓**	**✓**			[522]
Suppression of Tumorigenicity 2 (ST2)	Myocardial infarction		**✓**			**✓**	[524,525,526,527]
	Heart failure		**✓**			**✓**	[526,527,528]
	Aortic valve impairments		**✓**			**✓**	[529]
Growth Differentiation Factor 15 (GDF-15)	Myocardial infarction		**✓**				[373]
	Heart failure		**✓**	**✓**		**✓**	[375,390]
	Atrial fibrillation		**✓**	**✓**		**✓**	[395]
	Multiple CVDs		**✓**	**✓**			[7,378,380,385,386,390,391,396,398,399,400]
Endothelin-1 (ET-1)	Myocardial infarction		**✓**			**✓**	[267,429,430]
	Heart failure		**✓**	**✓**	**✓**	**✓**	[267,429,430]
	Multiple CVDs		**✓**	**✓**			[307,411,413,414,423,424]
Cytokines	Atrial fibrillation (interleukin-6, tumor necrosis factor-α and intercellular adhesion molecule-1)		**✓**		**✓**	**✓**	[467,468]
	Multiple CVDs		**✓**			**✓**	[530,531,532,533]
Lipoprotein-Associated Phospholipase A2 (Lp-PLA2)	Multiple CVDs		**✓**			**✓**	[447,448,534]
Soluble CD40 Ligand	Multiple CVDs		**✓**			**✓**	[453,535,536]
Serum Amyloid A	Multiple CVDs		**✓**			**✓**	[449,537]
Osteoprotegerin (OPG)	Multiple CVDs		**✓**			**✓**	[307,434,461,462]
Myeloperoxidase	Multiple CVDs		**✓**			**✓**	[63,359,504]
C-reactive protein (CRP)	Multiple CVDs		**✓**	**✓**	**✓**	**✓**	[467,533,537,538,539,540,541,542,543,544,545,546,547,548]
Erythrocyte sedimentation rate (ESR)	Multiple CVDs		**✓**			**✓**	[549]
Neutrophils and monocytes	Multiple CVDs		**✓**			**✓**	[463]
Cystatin C	Multiple CVDs		**✓**			**✓**	[550,551,552]

S: Screening, D: Diagnosis, P: Prognosis. (?) = biomarker is recommended to use for this purpose in some reports.

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
