# Peer review of "Cardiovascular Biomarkers: Lessons of the Past and Prospects for the Future"

_ijms, 2022, doi:10.3390/ijms23105680_

Round 1

Reviewer 1 Report

The authors described the involvement of cardiovascular diseases (CVDs) biomarkers in diagnostic, prognostic or screening purposes.

Overall, the review is well written. The topic is important and has impact their implementation in routine clinical practice.

The Table 1 contained a lot of data and somewhat confused. It would be better to summarize more briefly. On the other hand, authors should add a description about arrhythmia and cytokines in the Table.

Author Response

Editor Comments: We sent you a revision request earlier and would like to mention that the subtitles are not in numerical order. Please kindly revise your paper on this issue meanwhile.

Reply: Thank you for this comment. As requested, we have added the subtitles in numerical order in the revised manuscript. 

Reviewer-1

Reviewer-1: The authors described the involvement of cardiovascular diseases (CVDs) biomarkers in diagnostic, prognostic or screening purposes. Overall, the review is well written. The topic is important and has impact their implementation in routine clinical practice.

Reply: We would like to sincerely thank the Editor and Reviewer(s) for the time dedicated to reviewing our work, and for these supportive comments

Reviewer-1: The Table 1 contained a lot of data and somewhat confused. It would be better to summarize more briefly.

Reply: Thank you for this suggestion. As suggested, we have added a legend in Table 1 that summarizes more briefly this table (please see lines 721-736 of the revised manuscript).

Reviewer-1: On the other hand, authors should add a description about arrhythmia and cytokines in the Table.

Reply: Thank you for this suggestion. As suggested, we have added a description about arrhythmia and cytokines in the revised legend of Table 1 (please see lines 727-731 of the revised manuscript and added references [466], [467], [468], [469], [470]).

 Reviewer 2 Report

Cardiovascular disease (CVD) is a main cause of death worldwide. The National Institute of Health Consortium defined a biomarker as a “characteristic that is objectively measured and evaluated as an indicator of normal biological processes, pathogenic processes, or pharmacologic responses to a therapeutic intervention” . However, the American Heart Association outlined the extensive criteria for how newer biomarkers should be evaluated in a standardized fashion before their clinical use can be recommended in 2009. This is an interesting study. The following comment should be taken into account in order to improve the overall quality and readability of the manuscript.

Minor concern:

The association of heart failure classification and current and prospective biomarkers should be described more detail.

Author Response

Reviewer-2

Reviewer-2: Cardiovascular disease (CVD) is a main cause of death worldwide. The National Institute of Health Consortium defined a biomarker as a “characteristic that is objectively measured and evaluated as an indicator of normal biological processes, pathogenic processes, or pharmacologic responses to a therapeutic intervention”.

Reply: Thank you for this comment. We have added this in the introduction of the revised manuscript (please see lines 44-47 of the revised manuscript and added references [4], [5]).

Reviewer-2: However, the American Heart Association outlined the extensive criteria for however biomarkers should be evaluated in a standardized fashion before their clinical use can be recommended in 2009.

Reply: Thank you for this comment. We have added this in the introduction of the revised manuscript (please see lines 51-54 of the revised manuscript and added references [7]).

Reviewer-2: This is an interesting study. The following comment should be taken into account in order to improve the overall quality and readability of the manuscript. Minor concern: The association of heart failure classification and current and prospective biomarkers should be described more detail.

Reply: Thank you for this comment. Accordingly, we have added a description of this point in the revised conclusion section of the manuscript (please see lines 693-700 of the revised manuscript and added references [464], [241], [264]).